# Analysis of the Factors Intervening in the Prehospital Time in a Stroke Code

**DOI:** 10.3390/jpm13101519

**Published:** 2023-10-23

**Authors:** Álvaro Astasio-Picado, Yolanda Cruz Chueca, Miriam López-Sánchez, Rocio Ruiz Lozano, María Teresa González-Chapado, Vanesa Ortega-Trancón

**Affiliations:** 1Physiotherapy, Nursing and Physiology Department, Faculty of Health Sciences, University of Castilla-La Mancha, 45600 Toledo, Spain; yoliicruuz@gmail.com; 2Extremadura Health Service, 10300 Cáceres, Spain; miri-due@hotmail.es (M.L.-S.); rociorulo@gmail.com (R.R.L.); 3Nursing Department, Universitat Oberta de Catalunya, 08035 Barcelona, Spain; mteresagch@gmail.com (M.T.G.-C.); vortega1980@hotmail.com (V.O.-T.)

**Keywords:** time to treatment, emergency medical services, ictus, cerebrovascular accident

## Abstract

Introduction: Strokes continue to be considered public health problems due to the great social and health impact they entail. They are the second cause of death in the world, with a high incidence and prevalence. They are time-dependent diseases, and more than 80% of cases could be avoidable with greater management of risk factors. Objective: to analyze the factors that influence prehospital time in a stroke code. Assess the population’s knowledge of stroke symptoms and teach them how to act when a case is suspected. Document the continued training of health professionals for the early identification of patients with a suspected stroke. Demonstrate the importance of calling EMS as the first contact to reduce delays in prehospital time in a stroke. Methodology: A bibliographic review was carried out focusing on articles published between December 2014 and August 2023. The following databases were consulted: Pubmed (Medline), Dialnet, Google Scholar, Web of Science (WOS), Scielo, Scopus, and ScienceDirect. Results: After applying the article selection criteria and evaluating the quality of the methodology, a total of 18 articles were obtained. The results affirm that the importance of achieving a reduction in prehospital time is based mainly on knowledge of the symptoms and the use of new technologies. Conclusions: The evidence supports that the prehospital time of action in the stroke code is affected by numerous factors. These factors are determining factors in the time of action to achieve good effectiveness in the treatment of the pathology.

## 1. Introduction

Cerebrovascular diseases are pathologies of vascular origin that affect the brain, manifesting as strokes [1]. A stroke is considered a public health problem since it has a great social and health impact due to its notable incidence and prevalence. Furthermore, it is considered not only a health burden but also a family and personal burden due to the great life change suffered by people who suffer from it and its impact on their environment [2]. This pathology is the first cause of disability in adults and the second cause of dementia after Alzheimer’s. Furthermore, it is the second cause of death worldwide, the first cause of mortality in women, and the third in men [2]. Both incidence and prevalence increase with increasing age. It is estimated that there are around 71,780 new cases of strokes in Spain. Currently, around 27 thousand people die in Spain due to strokes, and it is believed that these figures may increase up to 40% [2]. After a stroke, patients usually suffer very disabling sequelae that deteriorate their quality of life, which entails a need for care and consumption of resources that entails high healthcare costs. It is estimated that the total healthcare cost of strokes is EUR 2908 million [2]. More than 80% of strokes are preventable, and the incidence can be reduced with exhaustive control of risk factors. Atrial fibrillation multiplies the risk of suffering a stroke by five, and they are also more serious and have a higher mortality [2]. A stroke is considered a medical emergency, which is why it must be treated in specialized units, such as stroke units. It requires early diagnosis and treatment since it is a time-dependent disease, and millions of neurons die every minute [3]. It must be taken into account that “time is the brain”, which is why good coordination between the different links in the healthcare chain is very important. This has led to the implementation of the stroke code, as well as knowledge on the part of the population of stroke symptoms for the early identification of the pathology [3].

### 1.1. Concept and Classification of the Stroke Code

The stroke code is a care procedure of a delimited territorial area that guarantees access wherever the person is being treated with cerebral reperfusion therapies, as long as they meet the inclusion criteria of the stroke code [4,5]. Its main objective is to reduce the time of action between the onset of symptoms and access to diagnosis and treatment and thus achieve the minimum possible consequences in recovery and increase the number of patients treated in stroke units. There are three types of stroke codes. The out-of-hospital stroke code consists of the rapid identification of the patient with stroke symptoms for transfer to the reference hospital [5]. The in-hospital stroke code is carried out while the patient is transferred to the hospital and consists of the activation of the diagnosis and the preparation of medical care by the neurologist at the reference hospital [5]. The interhospital stroke code is activated for the transport of a patient from a hospital that is not capable of administering cerebral reperfusion therapy to another hospital that is capable [5]. Citizens can access the public system through the 112 telephone number and the health center during hospital emergencies or hospitalization. The stroke code is responsible for guaranteeing access to reperfusion therapies through any entry modality to the comprehensive emergency system [5]. Through any of the above access modalities, both healthcare and non-healthcare professionals must be able to identify stroke symptoms early in order to activate the stroke code and begin to act as soon as possible [5]. To activate the stroke code, the patient must meet a series of criteria. These criteria are independence for the ABVD, time of onset of symptoms less than 8 h, and neurological alteration at the time of diagnosis, including weakness or paralysis in one hemibody, dysarthria, sudden headache without apparent cause, or difficulty walking [6]. Stroke units are organizations of groups of coordinated specialists specifically trained in the diagnosis and treatment of stroke, and they operate with specific protocols. The reduction in mortality of stroke patients who have been treated within these stroke units has been demonstrated [5].

### 1.2. Concept and Classification of Strokes

Cerebrovascular diseases are caused by alterations in the brain circuit that affect the functioning of the brain. It is very important to know the cause of the disease in order to carry out adequate treatment and effective secondary prevention to avoid a new episode. Therefore, there are multiple types of strokes. Depending on the etiology of the stroke, these are cerebral ischemia and hemorrhage [3]. Cerebral ischemia is caused by the arrest of blood circulation in the brain or part of the brain due to thrombosis or embolism. It is the most frequent since it causes 85% of cases [6,7]. According to the GEECV/SEN, it is classified into global cerebral ischemia and focal cerebral ischemia [3]. Within focal cerebral ischemia, only one area of the brain is affected. Depending on the duration, two types can be distinguished: transient ischemic attack and cerebral infarction or ischemic stroke. Within the latter, we can distinguish between atherothrombotic infarction, cardioembolic infarction, lacunar infarction, infarction of rare cause, and infarction of undetermined etiology [3]. However, hemorrhagic stroke is caused by extravasation of blood due to the rupture of a cerebral blood vessel [3]. It accounts for 15% of cases [7]. Depending on the location, shape, size, and etiology of the bleeding, they are classified into intracerebral hemorrhage and subarachnoid hemorrhage [3].

### 1.3. Stroke Treatment

Once the type of stroke has been diagnosed, treatment begins as soon as possible. There are different treatments for strokes, but the most used are reperfusion treatments and neuroprotection. Reperfusion treatment is a procedure through which the obstruction of the damaged vessel is removed, restoring blood flow [8,9]. It is the most effective treatment for ischemic strokes. It can be performed in two ways [10,11], Including intravenous thrombolysis, whose objective is to break up the clot that is blocking the artery. The medication used for this task is rTPA intravenously. The dose to administer is 0.9 mg/kg without ever exceeding 90 mg. 10% is administered as a bolus to observe that there are no allergies and once verified, reperfusion is started on a pump for 1 h. Always before starting this treatment, a neuroradiological study must be performed to rule out possible exclusion criteria [10,11]. On the other hand, endovascular or mechanical thrombectomy is an intravascular procedure used to remove the clot through an endovascular catheter [10,11,12,13]. Regarding neuroprotective treatment, there are no conclusive results; therefore, in clinical practice, there is no recommendation for neuroprotective drugs [5].

The objective of this study is to analyze the factors that influence prehospital time in a stroke code. Analyze the population’s knowledge of stroke symptoms and teach them how to act when a case is suspected. Verify the continued training of health professionals for the early identification of patients with a suspected stroke. Analyze the importance of calling EMS as the first contact to reduce delays in prehospital when a stroke happens.

## 2. Materials and Methods

The preparation of this work was carried out through a systematic bibliographic review of the articles found through searches in the following databases: Pubmed/Medline, Dialnet, Google Scholar, Web of Science (WOS), Scielo, Scopus, and ScienceDirect. To find the best possible scientific evidence, a series of inclusion and exclusion criteria were applied.

### 2.1. Information Sources and Search Strategy

The keywords for this review are time to treatment, emergency medical services, ictus, and cerebrovascular accident. To carry out the bibliographic search, different keywords in English were used, such as “Time-to-Treatment”, “Emergency Medical Services”, and “Stroke”. These have been validated by the DeCS and MeSH. Once selected, the corresponding Boolean operators were used: AND, as well as the necessary parentheses and quotation marks. The final search string is as follows: (“Time-to-Treatment”) AND “Emergency Medical Services”) AND “Stroke”.

### 2.2. Inclusion Criteria and Exclusion Criteria

The criteria that were taken into account for the selection of the relevant studies were the following. Inclusion criteria: the period between 2011 and 2023; article type: article review and article research; field: medicine; English language; sample in the adult population; and studies that provide scientific evidence justified by the level of indexing of articles in journals according to the latest certainties. Exclusion criteria: articles prior to 2011; language: not English; studies in which the population was minors; and studies that do not provide scientific evidence justified by the level of indexing of articles in journals according to the latest certainties.

### 2.3. Methodological Evaluation of the Data Used

For the methodological evaluation of the individual studies and the detection of possible biases, the evaluation was carried out using the PEDro Evaluation Scale. This scale consists of 11 items, providing one point for each element that is fulfilled. The articles that obtained a score of 9–10 points have an excellent quality, those between 6 and 8 points have a good quality, those that obtained 4–5 points have an intermediate quality, and, finally, those articles that obtained less than 4 points have a poor methodological quality.

The Scottish Intercollegiate Guidelines Network classification was used in the data analysis and assessment of the levels of evidence, which focused on the quantitative analysis of systematic reviews and the reduction in systematic error. Although it took into account the quality of the methodology, it did not assess the scientific or technological reality of the recommendations.

## 3. Results

The research question was constructed following the PICO format (Population/Patient, Intervention, Comparator, and Outcomes/Outcomes). It was detailed as P (Patients): adults of both sexes who have suffered a stroke; I (Intervention): analyze what factors influence the prehospital response time in the stroke code. C (Comparison): scientific evidence available; O (Results): check how they influence the action time (Figure 1).

Below is a table that shows the search strategy used to select the eighteen articles selected from the seven databases following the criteria of the identified studies, duplicate studies, title, abstract, full text, and valid studies of a definitive nature. The total number of valid articles is summarized in Appendix A.

The main difference between urban and rural environments is based on the transportation time of EMS. This is divided into the following time intervals: total time (from dispatch to arrival at the hospital), response time (from dispatch to arrival at the scene), time on the scene (from arrival at the scene to departure), and travel or transportation time (from departure from the scene to arrival at the hospital). Treatment time is the average total time elapsed between the start of symptoms and the onset of treatment. Door-to-needle time is the time between presentation at the hospital or ambulance and administration of fibrinolytic treatment (Table 1) [14,15].

Regarding the knowledge of the symptoms by health professionals, in the study carried out by Gorchs-Molist, the effectiveness of training health professionals was demonstrated. In 2014, only 55.2% of healthcare professionals were able to recognize a stroke. However, in 2018, after training, 85.5% of health professionals were able to recognize a stroke [16].

Regarding the knowledge of symptoms by patients, the studies carried out by Seo, Faiz, and Naguib compare the knowledge that the general population has about the symptoms that appear during an episode of stroke. It can be seen that this knowledge is very low in the population, which delays the early identification of the condition and late diagnosis and treatment (Figure 1) [17,18,19].

Regarding the activation of emergency medical systems as first contact, several studies analyze whether patients activate EMS as their first option when faced with a stroke episode or choose other means of transportation to get to the hospital. The time of the stroke care process is reduced if EMS are activated as the first option, although the vast majority of the population chooses other means of transportation (Table 2) [20,21,22,23].

Regarding prior notification by emergency medical services, prior notification to hospitals by EMS is recommended, as studies show it reduces delays in time-dependent therapies. However, non-notification lengthens those times that are crucial in this type of time-dependent pathology (Table 3) [24,25].

Regarding the importance of medical dispatchers, the study carried out by Cáceres compares the identification of strokes by medical dispatchers or by other health professionals. Medical dispatchers are responsible for transmitting calls to produce EMS deployment. Therefore, the identification of strokes by medical dispatchers significantly reduces the response times for said pathology (Figure 2) [26].

Regarding the use of telestroke consultations, in Al Kasab’s study, it can be observed how the use of telestroke consultations shortens the needle gate time; that is, the time between the diagnosis of the pathology and the administration of the treatment. Using TEMS (telestroke consultation in the emergency medical services unit) has a needle gate time of 21 min and using STS (standard telestroke consultation) has a needle gate time of 38 min [27].

Regarding mobile stroke units, various studies show that the activation of mobile stroke units reduces prehospital times for code strokes. This is because the patient receives early specialized care, which accelerates the patient’s entire care process, and receives a diagnosis in the shortest possible time so that fibrinolytic treatment can be administered (Table 4) [28,29].

Regarding the use of a mobile application to detect strokes, the use of a “Stop Stroke” mobile application has been shown to reduce stroke code action times, such as scene-to-destination time, CT gate time, and needle gate time; however, its use is still under investigation (Figure 3) [30].

Regarding the factors that influence the prehospital response time in the stroke code, prehospital response time is affected by numerous factors, which can reduce response times or increase them. Since it is a time-dependent pathology, it is important to act on the factors that increase prehospital times in order to reduce them, whenever possible (Table 5) [31].

## 4. Discussion

The studies reviewed in this research work provide information on the different factors to control the prehospital management of strokes.

Strokes continue to be a public health problem due to their high incidence and prevalence. There are numerous sequelae that patients can suffer after suffering a stroke, deteriorating their quality of life; therefore, early diagnosis and treatment are very important. They would be achieved with greater control over the factors that influence the prehospital time of the stroke code. One of the main problems in prehospital delays is the geographical area where the episode occurs. According to the article by Golden and Odoi, there are great differences between rural and urban environments. Only 2% of cases in urban areas would exceed the time guidelines from when the case is notified until arrival at the hospital, compared to almost 17% in rural areas [14]. The study carried out by Varjoranta also confirms these differences since it demonstrates how the time from arrival at the hospital to receiving fibrinolytic treatment has an average of 130 min in urban settings. However, in rural areas, it can reach 194 min, which increases the risk of not being able to receive fibrinolytic treatment due to exceeding the time limit to start treatment. Therefore, in rural areas, there is a longer prehospital delay time than in urban areas [15].

To reduce response times, knowledge of the symptoms by both health professionals and patients is very important. The work carried out by Gorchs-Molist demonstrates how good training of health professionals increases the recognition of up to 35% of strokes in the shortest possible time [16]. The articles by Faiz, Naguib, Soto-Cámara, and Seo show how the level of knowledge of the symptoms by the population continues to be very low, which leads to delays in prehospital time [17,18,19,31]. This can translate into the impossibility of carrying out fibrinolytic treatment, lengthening treatment times, and an increase in sequelae and morbidity. Furthermore, if the population does not recognize the symptoms, there is a delay in the activation of the EMS, lengthening the response times. Therefore, it is very important to develop health education programs for both the population and health professionals in order to reduce these times, placing special emphasis on the elderly population who are those most at risk of suffering a stroke. For this reason, age and educational level are also factors to take into account when determining strokes in the population. The studies carried out by Naguib, Soto-Cámara, and García Ruiz demonstrate how the young population and higher educational levels are associated with greater knowledge of stroke symptoms. Therefore, EMS activation occurs in less time, which reduces prehospital delays [19,22,31].

The severity of symptoms also influences the early recognition of a stroke case. The greater the severity of the symptoms, the greater the recognition of stroke, which translates into a reduction in response times, according to the articles by Soto-Cámara and Seo [17,31]. Another factor to take into account is whether the patient was alone or accompanied during the stroke episode. It has been demonstrated in the work carried out by Gonzalez-Aquines, Soto-Cámara, García Ruiz, and Seo that being in a public place or having witnesses to a stroke episode reduces response times [17,21,22,31]. On the contrary, living alone or being alone at the time of the episode increases the prehospital delay.

The activation of EMS as the first contact is essential to reduce prehospital delays. However, the majority of the population chooses other means of transportation, goes to the PC first, or decides to wait for the symptoms to pass. This leads to an increase in response times and, therefore, an increase in risks. Therefore, the importance of good population education about strokes and how to act is once again demonstrated, as reflected in the articles prepared by Alabdali, Gonzalez-Aquines, García Ruiz, Puolakka, and Soto-Cámara [20,21,22,23,31]. Furthermore, PN to EMS has shown a reduction in response times, according to the articles by McKinney and Kim [24,25]. The cases of patients with a suspected stroke arriving at the hospital without prior notice were high, which increased evaluation, diagnostic tests, and treatment times. Likewise, it has been proven that stroke cases identified by medical dispatchers reduce stroke response times compared to those identified by other professionals, according to the study carried out by Cáceres [26].

Likewise, the study by Al Kasab and Sami et al. demonstrates how the use of telestroke consultations has shown a reduction in the needle-to-door time of almost half compared to standard stroke consultations. In this way, patients are evaluated via videoconference by stroke experts and are thus transported directly to the appropriate healthcare center, avoiding time delays and avoiding delays due to referral to another center [27]. Likewise, in the articles by Cooley, S Regan et al. and Belt, Gary H et al., it is confirmed that the activation of the mobile stroke units reduces the response times by more than half compared to the non-activation of these units [28,29]. Therefore, in the case of a suspected stroke, the first thing to do is notify these mobile units. However, there are few mobile units available, since they are expensive and require specialized professionals. The article carried out by Andrew, Benjamin Y et al. discusses a mobile application used by EMS called “Stop Stroke”, which demonstrated the reduction in stroke response times [30]. In this way, the number of patients who can be treated with thrombolytic therapy increases.

As for the limitations of this study, although the results obtained are conclusive in response to the objectives of this study, larger samples could yield more conclusive results. The heterogeneity between studies means that the results found should be taken with caution. Given the scarcity of published clinical trials, it is difficult to address and see how these techniques affect patients holistically, which justifies future research.

## 5. Conclusions

In relation to the main objective of analyzing the factors that influence the prehospital time in a stroke code, it has been shown that the prehospital time of action in the stroke code varies according to numerous factors. These factors are associated with a reduction in prehospital time and include the early recognition of symptoms, activation of EMS as a first contact, not being alone at the time of the episode, greater severity of the stroke, the young population having greater knowledge of symptoms, people with a higher educational levels, living in an urban environment, and the use of telestroke consultations and mobile stroke units. However, not recognizing stroke symptoms, going first to the PC, not having witnesses at the time of the episode, or the elderly population not being aware of the symptoms are associated with an increase in prehospital time. Also, people with a low educational level, those living in rural areas, and a referral to another health center are associated with an increase in prehospital time. In relation to the objective of assessing the population’s knowledge of stroke symptoms and teaching them how to act when a case is suspected, the high degree of lack of knowledge among the population about the symptoms that appear in the event of a stroke episode is confirmed. This circumstance implies an increase in the prehospital time of action of the stroke since by not recognizing the symptoms, they stay at home waiting for them to pass or go to the PC, which decreases the probability of being able to receive fibrinolytic treatment. It is, therefore, essential to develop community awareness campaigns so that the population can recognize the symptoms of a stroke and, above all, learn how to act in the event of an episode. In accordance with the objective of documenting the continued training of health professionals for the early identification of patients with suspected stroke, it has been verified that continuing the training of health professionals on the stroke code protocol increases early recognition and, therefore, prehospital intervention time is reduced. Regarding the objective of demonstrating the importance of calling EMS as the first contact to reduce delays in prehospital time in a stroke, it has been proven that calling EMS is crucial to accelerate the entire prehospital care process and thus increase the percentage of patients receiving fibrinolytic treatment, reducing sequelae and morbidity. The importance of good health education for the population to know how to act if a stroke is suspected is once again emphasized.

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
