# Peer review of "Analysis of the Factors Intervening in the Prehospital Time in a Stroke Code"

_jpm, 2023, doi:10.3390/jpm13101519_

Round 1

Reviewer 1 Report

Dear authors,

thank you for this comprehensive work. However I would have some suggestions to correct:

1. I would suggest to consult a native speaker for correction of english grammar/terms

2. part 1.2. - classification - should be corrected:

a) "...cause that "produces"..." - meaning stroke, more suitable expression would be "ethiology of stroke", not "cause that produces stroke"

b) TIA is not a stroke, but it can be described in this part, however, the difference is not (only) the duration but mainly presence (or not) of morphological changes (lesions) of the brain 

c) in hemorrhagic stroke difference between ICH and SAH is not mainly in "location, shape and size" but also in etiology of bleeding

3. Therapy part 1.3. - resolving of the clot is not achieved only with rTPA, also I would suggest to skip detailed description of applying this therapy 

4. Table 1 - please explain the term "treatment time"? The term "door to needle" is properly used, but could you please add (or explain why is it missing) a usual "door to groin" time

5. Please make conclusions more concise by just clearly listing which factors you consider as main for delay/or decrease of pre-hospital time

Please see comment Nr 1

Author Response

Dear reviewer,
We appreciate the valuable time you have spent evaluating our manuscript. This assessment will make the article more valuable and have better scientific quality.
Regarding your recommendations:
1. I would suggest consulting a native speaker to correct English grammar/terms: We have sent the manuscript to the language department of the college to correct errors.
2. Part 1.2. - classification - needs to be corrected:
a) "...cause that "produces"..." - i.e. stroke, the most appropriate expression would be "etiology of stroke", not "cause that produces stroke": paragraph modified and marked in yellow .
b) TIA is not a stroke, but it can be described in this part, however, the difference is not (only) the duration but mainly the presence (or not) of morphological changes (lesions) of the brain. We appreciate the appreciation.
c) in hemorrhagic stroke the difference between ICH and SAH is not mainly in "location, shape and size", but also in the etiology of bleeding: paragraph modified and marked in yellow.
3. Therapy part 1.3. - clot resolution is not achieved with rTPA alone, I would also suggest omitting the detailed description of the application of this therapy: we do not agree with suppressing that information, but if the reviewer considers it appropriate we eliminate it.
4. Table 1: Explain the term “treatment time”. The term "door to needle" is used correctly, but could you add (or explain why it is missing) a common "door to groin" time?: Treatment time is the average total time elapsed between the start of symptoms and onset
treatment. Paragraph modified and marked in yellow.
5. Make the conclusions more concise by clearly listing which factors you consider to be the main factors for delaying or decreasing pre-hospital time: the conclusions have been synthesized.

Regarding the evaluation of the translation, the manuscript has been sent to the language department of the Faculty.

We are very grateful for your assessment and work. We hope that the article is to your liking for acceptance.

Reviewer 2 Report

The manuscript titled "Analysis of the Factors Intervening in the Prehospital Time in a Stroke Code" discusses various aspects related to stroke, a significant public health issue. The authors aim to analyze the factors influencing prehospital response times in stroke cases, assess the population's knowledge of stroke symptoms, the continued training of health professionals, and the importance of calling emergency medical services (EMS) as the first contact for reducing delays in prehospital stroke treatment. This work should be of wide interests to most researchers on neuroscience and biomedicines. 

This manuscript provides valuable insights into the factors affecting prehospital time in stroke codes. It offers a well-structured analysis, supported by relevant data. However, it could benefit from a more critical evaluation of the limitations and potential biases in the included studies. Additionally, recommendations for future research or practical implications for healthcare systems could enhance the manuscript's value.

The quality of English Language is high.

Author Response

Dear reviewer,
We appreciate the valuable time you have spent evaluating our manuscript. This assessment will make the article have greater value and scientific quality.
Regarding your recommendations:
a) We share with you the appreciation that this work is of great interest to the majority of researchers in neuroscience and biomedicine.
b) (...) the limitations and possible biases of the included studies. Furthermore, recommendations for future research or practical implications for health systems could increase the value of the manuscript: we have incorporated a limitations and recommendations for future studies section in the discussion section.

Regarding the evaluation of the translation, the manuscript has been sent to the language department of the Faculty.

We are very grateful for your assessment and work. We hope that the article is to your liking for acceptance.

Round 2

Reviewer 1 Report

Thank you for accepting the corrections.

According to my opinion, manuscript can be accepted after the previewed minor  corrections of English language.

Author Response

Dear reviewer,

We appreciate the valuable time you have spent evaluating our manuscript. This assessment will make the article more valuable and have better scientific quality.

Regarding the evaluation of the translation, the manuscript has been sent to the language department of the Faculty.

We are very grateful for your assessment and work. We hope that the article is to your liking for acceptance.